



# Air–surface exchange of gaseous mercury over permafrost soil: an investigation at a high-altitude (4700 m a.s.l.) and remote site in the central Qinghai-Tibet Plateau

Zhijia Ci[1], Fei Peng[2], Xian Xue[2], and Xiaoshan Zhang[1]

[1]Research Center for Eco-Environmental Sciences, Chinese Academy of Sciences, Beijing, 100085, China

[2]Cold and Arid Regions Environmental and Engineering Research Institute, Chinese Academy of Sciences, Lanzhou, 730000, China

*Correspondence to*: Z. J. Ci (zjci@rcees.ac.cn)

**Abstract.** The pattern of air–surface gaseous mercury (mainly Hg(0)) exchange in the Qinghai-Tibet Plateau (QTP) may be unique because this region is characterized by low temperature, great temperature variation, intensive solar radiation, and pronounced freeze-thaw process of permafrost soils. However, air–surface Hg(0) flux in the QTP is poorly investigated. In this study, we performed filed measurements and controlled field experiments with dynamic flux chambers technique to examine the flux, temporal variation and influencing factors of air–surface Hg(0) exchange at a high-altitude (4700 m a.s.l.) and remote site in the central QTP. The results of field measurements showed that surface soils were net emission source of Hg(0) in the entire study. Hg(0) flux showed remarkable seasonality with net high emission in the warm campaigns and net low deposition in winter campaign, and also showed the diurnal pattern with emission in daytime and deposition in nighttime, especially on days without precipitation. Rainfall events on the dry soils induced large and immediate increase in Hg(0) emission. Snowfall events did not induce the pulse of Hg(0) emission, but snow melt resulted in the immediate increase in Hg(0) emission. Daily Hg(0) fluxes on rainy or snowy days were higher than those of days without precipitation. Controlled field experiments suggested that water addition to dry soils significantly increased Hg(0) emission both in short and relatively long timescales, and also showed that UV radiation was primarily attributed to Hg(0) emission in the daytime. Our findings imply that a warm climate and environmental change could facilitate Hg release from the permafrost terrestrial ecosystem in the QTP.



# 1 Introduction

Soils represent the largest Hg reservoirs in ecosystems and play a major role in the global Hg cycle (Selin, 2009; Agnan et al., 2016). Background soils receive Hg input from atmospheric deposition, which is mainly retained in organic-rich layers of upper soils (Schuster, 1991; Khwaja et al., 2006). Under favorable conditions, Hg in soils can be reduced to Hg(0) and then emitted to the overlaying air because of its high volatility (Schlüter, 2006). Therefore, soils can serve as both sources and sinks of atmospheric Hg (Pirrone and Mason, 2009; Amos et al., 2013; Agnan et al., 2016).

In the past several decades, efforts have been made to improve the understanding of soil Hg biogeochemistry (Zhang and Lindberg, 1999; Lin et al., 2010; Schlüter, 2006; Jiskra et al., 2015). Measurements across various types of soils and climates show that air–soil Hg(0) exchange has highly spatial and temporal variation and bidirectional exchange behavior (Agnan et al., 2016 and references therein). Field measurements and laboratory experiments highlight that various factors and processes influence air–surface Hg(0) exchange, including concentrations and species of soil Hg (Gustin et al., 1999, 2002; Hintelmann et al., 2002; Bahlmann et al., 2006; Kocman and Horvat, 2010; Eckley et al., 2011; Edwards and Howard, 2013; Mazur et al., 2015), solar radiation (Gustin et al., 2002; Moore and Carpi, 2005; Gustin et al., 2006; Xin et al., 2007; Fu et al., 2008a; Kocman and Horvat, 2010; Park et al., 2014), precipitation (Lindberg et al., 1999; Gustin and Stamenkovic, 2005; Gabriel et al., 2011), soil temperature and moisture (Gustin et al., 1997; Gustin and Stamenkovic, 2005; Ericksen et al., 2006; Xin et al., 2007; Briggs and Gustin, 2013; Park et al., 2014; Mazur et al., 2015), soil organic matter and pH (Yang et al., 2007; Xin and Gustin, 2007; Mauclair et al., 2008); land cover (Dommergue et al., 2003; Ericksen et al., 2005; Cobbett et al., 2007; Gabriel and Williamson, 2008; Zhu et al., 2011; Durnford et al., 2012a, b; Toyota et al., 2014a, b); atmospheric Hg(0) concentrations and other chemical compositions (Engle et al., 2004; Xin and Gustin, 2007; Fu et al., 2008a), biological activity (Choi and Holsen, 2009), as well as atmospheric turbulence (Gustin et al., 1997; Poissant et al. 1999). Existing studies on Hg(0) dynamics at air–surface interface are mainly performed in temperate regions (Agnan et al., 2016 and reference therein). The seasonal frozen soils and permafrost widely distribute, accounting for almost 70% of terrestrial area of Earth (NSIDC). However, the knowledge of air–surface Hg(0)



dynamics in cold region is limited (Cobbett et al., 2007; Durnford and Dastoor, 2011). Most current
parameters of air–soil Hg(0) exchange applied in Hg biogeochemical models are mainly derived from
temperate regions of North America and Europe (Zhu et al., 2016).
The Qinghai-Tibet Plateau (QTP) is located in the western China with the area of 2.5 million km$^2$
and mean altitude of > 4000 m. Due to the high altitude and subsequent low temperature, a significant
portion (~ 1.5 million km$^{-2}$) of the QTP is underlain by permafrost (Kang et al., 2010). Because of the
harsh natural environment, limited research resources and difficulty of access and sampling logistics,
studies on Hg biogeochemistry in the QTP are limited. The role of QTP in the regional and global Hg
biogeochemical cycle is poorly understood (Ci et al., 2012; Agnan et al., 2016). At present, Hg studies
in the QTP mainly focused on the investigations of Hg concentration, speciation and distribution in
environmental samples, such as air (Fu et al., 2008b; 2012; Yin et al., 2015), snow and glacier (Loewen
et al., 2007; Wang et al., 2008; Zhang et al., 2012; Huang et al., 2012), and rain water (Huang et al.,
2013). The knowledge of Hg(0) dynamics at air–surface interface in the QTP is extremely poor. The
unique climatic condition, land cover and soil property suggest the need for the specific air–soil Hg(0)
flux data and mechanism representative of the environmental setting in the QTP to better constrain
global natural sources inventories (Ci et al., 2012; Agnan et al., 2016).
In this study, we applied dynamic flux chambers (DFCs) technique to investigate the flux,
temporal variation and influencing factors of air–surface Hg(0) exchange at a representative research
station in the central QTP. Meanwhile, controlled field experiments were performed to explore the
effect of rainfall and different wavebands of solar radiation on air–soil Hg(0) flux. Combining the result
of this study and other knowledge, we discuss the effect of future climatic and environmental change on
air–surface Hg(0) dynamics in the QTP.

## 2 Methods

### 2.1 Study site

The study was performed at the Beiluhe Permafrost Engineering and Environmental Research
Station affiliated to the Cold and Arid Regions Environmental and Engineering Research Institute,



Chinese Academy of Sciences (CAREER–CAS). The elevation of the Beiluhe region is about 4700 to
4800 m a.s.l.. The station ($34^o$ 49' 45'' N, $92^o$ 56' 06'' E) lies over the continuous permafrost zone in the
central QTP (Fig. 1). The terrain is undulation with sparse vegetation and surface fine sands or gravels.
The thickness of the active layer and permafrost around the station is 2.0–3.2 m and 60–200 m,
respectively; the active layer begins to freeze in September and thaw in May (Peng et al., 2015a). The
Beiluhe region experiences a continental climate with cold winter (up to –30 ℃) and warm summer (up
to 25 ℃). The magnitude of daily air temperature is up to 30 ℃, and the annual mean surface air
temperature is about –2 to –3 ℃ (Peng et al., 2015a). The solar radiation is high and characterized by
intense UV radiation (Wei et al., 2006). The mean annual precipitation is about 300 mm and mostly
occurs during the May to October under the influence of the Southern Asian Monsoon; the annual
potential evapotranspiration (~1300 mm) greatly exceeds the precipitation (Peng et al., 2015a). As a
remote region, there is no direct human activity to influence the local Hg cycle.
**2.2 Measurement of air–surface Hg(0) flux**

The dynamic flux chambers (DFCs) technique was widely used to investigate Hg(0) flux between

air–surface interface because it is inexpensive, portable, easy to set up and operate (e.g., Kim and
Lindberg, 1995; Carpi and Lindberg, 1998; Gustin et al., 2006; Wang et al., 2006; Dommergue et al.,
2007; Fu et al., 2008a; Kocman and Horvat, 2010; Edwards and Howard, 2013). The principle for
measuring air–surface Hg(0) flux using the DFCs technique involves placing a chamber over a surface
and measuring the difference in air Hg(0) concentrations at inside and outside of the chamber. DFC
means continuously draw ambient air into the chamber through inlets at a set flushing flow rate. Air–
surface Hg(0) flux was calculated using Eq. (1),
$$F = Q\frac{C_0 - C_i}{A} \quad \text{(Eq. 1)}$$
where $F$ is the Hg(0) flux (ng m$^{-2}$ h$^{-1}$), $Q$ is the flushing flow through the chamber (m$^3$ h$^{-1}$), $A$ is the
footprint of the chamber (m$^2$), $C_0$ and $C_i$ (ng m$^{-3}$) is air Hg(0) concentrations at outlet and inlet of the
chamber, respectively. Positive flux values indicate Hg(0) emission from the surface into the air;
negative flux values represent Hg(0) deposition to the surface from the air. The advantage and



limitation of DFCs technique for determining Hg(0) flux have been extensively discussed in previous

studies (e.g., Wallschläger et al., 1999; Gillis and Miller, 2000; Lindberg et al., 2002; Eckley et al.,

2010; Lin et al., 2012; Sommar et al., 2012; Zhu et al., 2015a, b).

In this study, quartz chambers were constructed for measuring Hg(0) flux and exploring the effect

of different rainfall depths and radiation condition on the Hg(0) flux. Quartz glass has many advantages

as construction material of chamber for determining Hg(0) flux in background soils. First, it has high

transmittance of the full spectrum of solar radiation, especially UV waveband (Fig. S1 in Supplement).

Therefore, quartz chamber is suitable to determine the more "actual" Hg(0) flux because the short

wavelength of solar radiation has been found to have important effect on Hg(0) dynamics at the air−soil

interface (Moore and Carpi, 2005; Bahlmann et al., 2006). Second, it has low potential for Hg(0)

adsorption and is easy to clean by heating to remove Hg bonding on the surface (Ci et al., 2016a). This

will decrease the systematic blank of measurement, which is critical for investigating Hg(0) flux over

background soils (Carpi and Lindberg, 1998).

Our semi-cylindrical quartz chamber was 8 cm high and 24 cm length with a footprint of 0.0384 m$^{-2}$ (0.16 m x 0.24 m) and an internal volume of 2.41 L, which is similar to previous studies (Eckley et al.,

2010 and reference therein). The chamber had nine inlets (8 mm in diameter) and three outlets which

were on the two opposite section of the chamber. The inlet sampling tube was placed near the ground

surface (3 cm above the surface) directly near the inlet of the chamber. A flushing ambient air was

drawn by vacuum pump (KNF, Inc. Germany) with 3.0 L min$^{-1}$ (0.18 m$^3$ h$^{-1}$) through the chamber.

Since the harsh environment condition and the unstable power supply, the usage of the commercial

automatic Hg analyzer (such as Tekran 2537) to conduct filed measurements of Hg(0) flux is

challenging in the Beiluhe station. Therefore, air Hg(0) concentrations in both inlet and outlet of the

chamber were monitored manually by gold trap simultaneously with a 2–3 h intervals (Ci et al., 2016b).

The air was pumped through gold trap using air pump (KNF, Inc. Germany) with 0.50 L min$^{-1}$ (0.03 m$^3$ h$^{-1}$). Hg(0) collected on gold traps was quantified on site by a cold vapor atomic fluorescence

spectrophotometer (CVAFS, Model III, Brooks Rand, USA) using two-stage gold amalgamation

method (Fitzgerald and Gill, 1979; Ci et al., 2011, 2016a). Gold trap efficiencies were determined in



laboratory (Hg(0) concentration: 3.2 to 13.4 ng m$^{-3}$) and field (Hg(0) concentration: ~1 to 2 ng m$^{-3}$ to
9.4 ng m$^{-3}$), and multiple measurements using gold traps in series showed no breakthrough at the sample
flow rate of 0.50 L min$^{-1}$ for 5 hours. The method detection limit was 0.03 ng m$^{-3}$ and precision was
$3 \pm 2\%$.

The turnover time obtained from this protocol was 0.68 min, which is similar to previous studies

(Eckley et al., 2010 and reference therein). The flow rates of air both the inlet and outlet of chamber
were adjusted by a needle valve and controlled by a rotameter. Prior to the measurement, the rotameters
were calibrated by a mass flow meter and a volumetric gas meter. The accuracy of flow rate was $\pm 3\%$.
The sampling control system was installed in the tent near the soil plot (<2 m).

In this study, a bare soil plot of 2 x 2 m was chosen and separated to four subplots (1 x 1 m, labeled

as Subplot A to D) to measure Hg(0) flux and further to investigate the effects of different wavelengths
of solar radiation on the Hg(0) exchange. Geologic unit of this soil plot is representative of major
lithologic units of the Beiluhe region. The study was conducted in four campaigns (May, September,
and December 2014 and May–June 2015), covering typical intra-annual meteorological condition in the
study region. It is well-known that the application of the chamber isolates the soil surface from air
turbulence and rain/dew/frost/snow, altering many environmental parameters that influence Hg(0)
production/consumption and deposition/emission at air–soil interface (Sommar et al., 2012 and
references therein). In the Beiluhe region, the rain/snow/dew/frost commonly occurs in short time scale
($10^1$–$10^2$ minutes) because of the high-altitude location and great variation of temperature and unstable
weather conditions. To minimize the effect of great variation of soil surface condition on Hg(0) flux
measurements, when only to measure Hg(0) flux, we used the same chamber to measure Hg(0) flux in
four subplots (Subplot A to D) in turn. For the rainy and snowy days, if possible, the chamber was
moved to next subplot generally after the rain or snow because studies confirmed the significant
influence of precipitation on Hg(0) flux in short time scale (Lindberg et al., 1999; Gustin and
Stamenkovic, 2005; Johnson et al., 2003; Song and Van Heyst, 2005; Lalonde et al., 2001, 2003;
Dommergue et al., 2003, 2007, 2012; Faïn et al., 2007; Bartels-Rausch et al., 2008; Brooks et al., 2008;
Mann et al., 2014, 2015). For the days without precipitation, the chamber was moved to next subplot
before sunrise to capture the effect of frost or dew on air–surface Hg(0) flux in the light.



In this study, all materials in contact with Hg(0) were quartz, Teflon or borosilicate glass. The
chambers and tubing were rigorously acid washed (Ci et al., 2016a). The quartz chambers were heated
to 650 ℃ for 2 h prior the measurement to further remove all Hg (Ci et al., 2016a). System blanks of
the four chamber systems were systemically inspected on site by placing acid-cleaned Teflon filter
beneath the chamber and routinely inspected before and after the measurements under the field
condition. The overall blank results were taken as the average of all chamber blanks for each particular
day or from the entire period if continuous monitoring was conducted. The blank values of the four
chambers were found to be very low (mean $\pm$SD: –0.02 $\pm$0.03 ng m$^{-2}$ h$^{-1}$; range: –0.20–0.07 ng m$^{-2}$ h$^{-1}$)
and were not significantly different (p>0.05) throughout the entire study. We also found that the mean
blank values were not significantly different (p>0.05) from zero and were insignificant compared to the
measured Hg(0) fluxes (see below), then Hg(0) flux data was not blank-corrected in this study.

**2.3 Determination of soil Hg**

Surface soil samples (0–2 cm) were collected from four soil subplots during June 2014 campaign.
Soil samples were freeze-dried and homogenized for total Hg determination using a Milestone's DMA
direct Hg analyzer (detection limit: 0.01 ng Hg or 0.15 ug kg$^{-1}$) following the EPA Method 7473
(Briggs and Gustin, 2013), and the analytical accuracy was 3%.

**2.4 Measurements of environmental variables**

A meteorological station that located 60 m from the soil plot was used to collect the following
environmental variables: air temperature ( ℃), relative humidity (%), wind speed (m s$^{-1}$), precipitation
(mm), photosynthetically active radiation (PAR, μ mol m$^{-2}$ s$^{-1}$), surface soil temperature ( ℃). The
surface soil temperature was monitored by an ASI-111 Precision Infrared Radiometer (Campbell
Scientific Inc. USA). This sensor was installed on a bar 0.8 m above the soil surface and provided a
non-contact measurement of the surface temperature. We also measured the soil temperature at 1.0 cm
soil depth at the inside of chamber, and no significant difference was found between the soil
temperature in the outside and inside of chamber. Details on the measurements of environmental
variables are given in Peng et al. (2015a). The thickness of snowpack was measured manually.

**3 Results and Discussion**



### 3.1 Soil Hg

Soil Hg concentrations of four subplots varied from $13.11 \pm 0.51$ to $12.83 \pm 0.81$ μg kg$^{-1}$, suggesting that the study region is a typical background soil for Hg. Soil Hg concentrations and air–soil Hg(0) flux of four subplots were not statistically different (p>0.05), indicating the properties of the soil plot were homogeneous.

### 3.2 Hg(0) in ambient air

Figure 2 shows the temporal variation of air Hg(0) concentrations inside and outside of chamber, air–surface Hg(0) flux and environmental variables during four campaigns in 2014–2015. Hg(0) concentrations of ambient air ranged from 0.93 to 1.78 ng m$^{-3}$ with a mean of $1.36 \pm 0.17$ ng m$^{-3}$ (n=361), which were slightly lower than those in typical Northern Hemisphere background region (~1.5 ng m$^{-3}$, Sprovieri et al., 2007; Ebinghaus et al., 2011). To our knowledge, four measurements (including this study) have been conducted to determine atmospheric Hg over the QTP (see Fig. 1 for the locations). The gradient of atmospheric Hg(0) over the QTP was characterized by high concentrations in Mt. Gongga ($3.98 \pm 1.62$ ng m$^{-3}$, Fu et al., 2008b) and Mt. Waliguan ($1.98 \pm 0.98$ ng m$^{-3}$, Fu et al., 2012), moderate concentrations in the Beiluhe ($1.36 \pm 0.17$ ng m$^{-3}$, this study), and low concentrations in the Nam Co ($0.96 \pm 0.19$ ng m$^{-3}$, Yin et al., 2015). It seems that the sampling station with relatively long distance from the source region of atmospheric Hg(0) (i.e., the Central China, Streets et al., 2005) had relatively low atmospheric Hg(0) concentrations (Fig. 1). Atmospheric Hg(0) concentrations were high in three warm campaigns and low in winter campaign in the Beiluhe station (Fig. 3), which is consistent with Mt. Gongga (Fu et al., 2008b) and Mt. Waliguan (Fu et al., 2012).

### 3.3 Air–surface Hg(0) flux and influencing factors

### 3.3.1 Temporal variation of Hg(0) flux

The mean of air–surface Hg(0) flux in the entire study period were 2.86 ng m$^{-2}$ h$^{-1}$ (25.05 μg m$^{-2}$ y$^{-1}$), indicating that surface soils were net emission source of Hg(0). Hg(0) flux in this study is comparable to those over background soils (–10–10 ng m$^{-2}$ h$^{-1}$, Wang et al., 2006; Ericksen et al., 2006; Fu et al., 2008a), but greatly lower than those over Hg-enriched soils ($10^2$–$10^3$ ng m$^{-2}$ h$^{-1}$, Gustin et al.,





1999; Wang et al., 2007; Edwards and Howard, 2013), indicating that soil Hg concentrations may be the
dominant factor for controlling the magnitude of Hg(0) emission flux (Agnan et al., 2016 and reference
therein).
Figure 2 shows that the Hg(0) flux was highly variable. The highest Hg(0) emission fluxes of 28.46
ng m$^{-2}$ h$^{-1}$ were observed at 13:00–15:00 on 13 September 2014 after a rain event on dry soils. The
highest Hg(0) deposition fluxes of −6.24 ng m$^{-2}$ h$^{-1}$ were observed at nighttime over the cold and dry
surface soil during December 2014 campaign (03:30–06:30, 24 December 2014).
Hg(0) flux generally showed the diurnal pattern with high emission in daytime and remarkable
deposition in nighttime, especially on days without precipitation (Fig. 2). Many studies have confirmed
that solar radiation is one of the most important drivers for soil Hg(0) emission (Xin and Gustin, 2007;
Choi and Holsen, 2009; Kocman and Horvat, 2010); high surface temperature also facilitates Hg(0)
production and subsequent emission (Park et al., 2014). Therefore, the two environmental variables
jointly regulate the diurnal pattern of Hg(0) flux. An in-depth discussion on synergistic effects of solar
radiation and surface temperature on Hg(0) flux is provided in below. Interestingly, the diurnal pattern
of Hg(0) flux of each day during December 2014 campaign was almost identical, which may be
associated with very similar weather conditions throughout the entire campaign.
Hg(0) flux showed pronounced seasonality with high emission in three warm campaigns and net
low deposition during winter campaign (Fig. 3). Similar seasonality has been reported in many other
studies (e.g., Gabriel et al., 2006). As discussed below, in warm seasons, some environmental variables,
such as high solar radiation, surface temperature and precipitation, facilitate the soil Hg(0) emission.

**3.3.2 Effect of precipitation (rain and snow) on air–surface Hg(0) flux**

Many studies addressed that precipitation greatly influences air–surface Hg(0) flux over different
timescales (Lindberg et al., 1999). In previous studies, investigators mainly focused on the effect of
rainfall/watering on air–soil Hg(0) flux (Lindberg et al., 1999; Johnson et al., 2003; Gustin and
Stamenkovic, 2005; Song and Van Heyst, 2005; Corbett-Hains et al., 2012) or the fate and transport of
Hg(0) at air–snow interface (Lalonde et al., 2001, 2003; Ferrari et al., 2005; Dommergue et al., 2003,
2007; Faïn et al., 2007; Bartels-Rausch et al., 2008; Brooks et al., 2008; Steen et al., 2009; Durnford et



al., 2012a, b; Mann et al., 2015). The field study on the effect of snowmelt on Hg(0) flux is very limited
(Cobbett et al., 2007). The precipitation at the Beiluhe station mainly occurs during May to October
(Peng et al., 2015a). Due to the high-altitude location of the Beiluhe station, snow event commonly
occurs in May to June and late September to October. Because of intensive solar radiation and surface
temperature, the snow melts or sublimates in short time scale ($10^0$–$10^2$ hour), i.e., little/no snow
accumulation occurs for long time (>3 day). Therefore, the Beiluhe region provides an unique
opportunity to investigate the different effects of rain, snow, and snowmelt on the air–surface Hg(0)
flux over different timescales.
Firstly, the effect of rain events on Hg(0) flux was investigated. Many studies reported that the
rainfall or watering of dry soils can promote soil Hg(0) emission immediately (Lindberg et al., 1999;
Gustin and Stamenkovic, 2005). We also found that the Hg(0) emission flux increased immediately
following the rainfall (Fig. 2). This finding is consistent with many other studies (Gustin and
Stamenkovic, 2005; Johnson et al., 2003; Lindberg et al., 1999; Song and Van Heyst, 2005).
Investigators supposed that the dramatic increases in Hg(0) emission may be attributed to the physical
displacement of Hg(0) present in soil air and desorption of loosely bound Hg(0) on soil particles by the
infiltrating water (Johnson et al., 2003; Gustin and Stamenkovic, 2005). Notably, Fig. 2 displays that the
pulse of Hg(0) emission after the rainfall was also observed at nighttime (such as 0:00 to 01:00 on 4
September 2014). Similar phenomenon was also documented by our controlled experiments (see below).
This indicates that the immediate increase in Hg(0) emission might not be controlled by photochemical
processes but by physical processes.
Soil moisture condition may also significantly regulate Hg(0) flux over relatively long timescales
(from hours to several days). Therefore, many experiments studied the effect of water addition on the
magnitude and pattern of air–soil Hg(0) flux over different timescales (Johnson et al., 2003; Gustin and
Stamenkovic, 2005). However, most of studies were performed in controlled laboratory or mesocosm
settings under certain well-defined, but not necessarily environment relevant conditions (Johnson et al.,
2003; Gustin et al., 2004; Gustin and Stamenkovic, 2005; Song and Van Heyst, 2005; Kocman and
Horvat, 2010; Corbett-Hains et al., 2012; Park et al., 2014). In this study, it was also challenging to
reveal the effect of rainfall on Hg(0) flux over relatively long timescales via field measurement since



the intermittent rain events occurred irregularly during June 2014 and September 2014 campaign (Fig.
2). For a better understanding of the effect of water addition, we chosen another similar soil plot with
homogeneous soil property to conduct controlled field experiments to explore the effect of different
rainfall depths on Hg(0) flux over different timescales (from minutes to hours). The controlled
experiments were performed during May–June 2015 campaign since this period had high surface
temperature and low precipitation (Fig. 2). In the Beiluhe station, hourly rainfall depth rarely exceeds
15 mm (Peng et al., 2015a). Therefore, we designed four different treatments of rainfall depth (0 mm, 1
mm, 5 mm, and 15 mm). The water addition to the dry soils commenced at night (01:40) on 30 May
2015 to exclude the effect of photochemical process in the first hours of experiments. We added the
Milli-Q water (Hg concentration < 0.2 ng L$^{-1}$) to the inside and outside of the chamber by pre-clean
plastic syringe within 10 min to simulate the three different rain depths. Four chambers were used to
simultaneously measure Hg(0) flux over these four treatments for 22 hours (from 01:00 to 23:00) with
the same protocol described in the Methods section. Hg(0) flux was measured with 20 min intervals in
the first hours (from 01:00 to 04:20) of the experiments to investigate the temporal variation of Hg(0)
flux in the short timescale, and with 1 hour intervals for the rest period of the experiments.

The high-time resolution measurements captured the immediate and dramatic increases in Hg(0)

emission flux after the watering of dry soils (Fig. 4). The baseline Hg(0) flux of 0 mm treatment was
used as the benchmark for the different rainfall depth treatments to be compared against. Obviously, the
higher amount of water addition resulted in longer duration and higher accumulative flux of Hg(0)
emission pulse. The duration of Hg(0) emission pulse for 1 mm and 5 mm treatment was < 20 min
(from 01:40 to 02:00) and ~40 min (from 01:40 to 02:20), respectively, which was lower than that of 15
mm treatment (~80 min, from 01:40 to 03:00). The duration of Hg(0) emission for 15 mm treatment in
the daytime was also longer than that of 1 mm and 5 mm treatment (Fig. 4).

As shown in Fig. 4, the cumulative flux of Hg(0) emission during the entire study period mainly

included two fractions: the pulse of Hg(0) emission after the watering (i.e., emission flux by watering),
and the Hg(0) emission during the daytime (i.e., emission flux by radiation). Figure 5 displays that both
"emission flux by watering" and "emission flux by radiation" for 15 mm treatment were significantly
higher than those of 1 mm and 5 mm treatment. As mentioned above, the dramatic increase in Hg(0)





emission after the simulated rain can be explained by physical displacement of interstitial soil air by the
infiltrating water. The long emission duration and large immediate emission flux for soil plot with high
water addition can be explained by that the more water needed longer time to percolate the soil column
and displaced more soil Hg(0). Many previous studies suggested that the magnitude of Hg(0) emission
with a rainfall or stimulated rain depended on soil moisture condition, i.e., if the amounts of water
received by the soils was less than needed to saturate, the soil surface showed an immediate increase in
Hg(0) emission; after the soil became saturated, Hg(0) emission from surface soil was suppressed
(Klusman and Webster, 1981; Lindberg et al., 1999; Johnson et al., 2003; Gustin and Stamenkovic,
2005). In this study, the pulse of Hg(0) emission flux for 15 mm treatment was significantly higher than
that of 5 mm and 1 mm treatment (Fig. 5). The field water capacity and bulk density of soil in the
Beiluhe region is about 28% and 1 g cm$^{-3}$ (Peng et al., 2015b), indicating that the 5 mm treatment may
induce the upper soil to saturate in short timescale since the duration of water addition was short (< 10
min). However, the pulse of Hg(0) emission for 15 mm treatment was significantly higher than that of 5
mm treatment. The surface soils with high sand content in the Beiluhe region have a high rate of water
infiltration and subsequently great infiltration depth. This process potentially increases the displacement
of soil Hg(0) and facilitates Hg(0) emission, as mentioned above. Therefore, in the field condition, the
duration and flux of pulse Hg(0) emission following water addition depends not only on how much
water received and soil moisture condition but also soil texture and soil water dynamics.
The water addition also increased the Hg(0) emission in the daytime, showing more water added,
longer duration of Hg(0) emission, and more Hg(0) emitted (Fig. 4 and Fig. 5). After the surface soil
was visibly dry, Hg(0) flux over soil plots with water addition had no significant difference from that of
the soil plot without water addition (i.e., 0 mm treatment). This result is consistent with many other
controlled studies. For example, Gustin and colleagues (Johnson et al., 2003; Gustin and Stamenkovic,
2005) found that once the soil water content became less than saturated, Hg(0) emission flux would be
significantly enhanced especially during the daytime, and once sufficient drying occured, the magnitude
of Hg(0) emission flux tended to gradually decrease. Investigators suggested that as the water
evaporates and soil dries, capillary action drives the upward movement of water and chemicals
(including Hg components) and recharge the Hg pool in surface soils (i.e., the "wick effect") and





subsequently favors the Hg(0) production and emission via photochemical processes in the light (Gustin and Stamenkovic, 2005). In our study, even for the wettest soil plot (i.e., 15 mm treatment), the surface soils were visually unsaturated in the daytime because of the low water retention, high infiltration rate of local soils and intensive solar radiation. Therefore, the pattern of Hg(0) emission for soil plots with high water addition is comparable to those of the above-mentioned studies.

Secondly, the effect of snow events on Hg(0) flux were investigated. One of the most significant differences between the rainfall and snowfall on the effect of Hg(0) exchange was that the snowfall did not induce the remarkable pulse of Hg(0) emission. For example, at 10:10 on 11 June 2014, a heavy snowfall occurred and continued to 11:20. The great thickness of snowpack reached to ~12 cm. However, no remarkable pulse of Hg(0) emission was observed during the snowfall. Instead, the Hg(0) dynamics at air–snow interface showed the clear diurnal pattern with high emission in daytime and deposition or emission albeit rather small in nighttime. This finding is consistent with previous studies on air–snow interface (Cobbett et al., 2007). It can be seen that the pattern of Hg(0) dynamics at air–snow interface was similar with that at air–soil interface, indicating that Hg(0) emission from surface snow was also mainly regulated by photochemical processes (Ferrari et al., 2005; Dommergue et al., 2003, 2007). However, it is well-known that the snowpack is a porous matrix, and gases are subjected to diffusion in the snowpack. Therefore, our measurements of Hg(0) flux at air–snow interface did not exclude the effect of Hg(0) dynamics at soil–snow interface, especially the low thickness (< 12 cm) of snowpack in the study.

We found that the snow melting led to the remarkable peak of Hg(0) emission. For example, during the sunrise of 12 June 2014, a precipitation with rain and snow induced the snowpack (12 cm) to melt suddenly and completely (i.e., the bare soil with no surface snow), a pulse of Hg(0) emission (~8 ng m$^{-2}$ h$^{-1}$) was observed, which was the largest Hg(0) emission flux during June 2014 campaign. We supposed that the great increase in Hg(0) emission by snowmelt in this study was consistent with the effect of rainfall, i.e., the displacement of soil Hg(0) during the snowmelt permeation of the soil column resulted in the dramatic increase in Hg(0) emission. At present, the study on the effect of snowpack melting on Hg(0) emission is limited. Cobbett et al. (2007) also found the remarkable increase in Hg(0) emission in Canadian Arctic during the snow melt, although the Hg(0) flux was relatively small.





Finally, the effect of precipitation (including rain and snow) on daily Hg(0) flux was investigated.
The above-mentioned results and discussion suggest that the precipitation has great potential to
facilitate soil Hg(0) emission over different timescales via physical, chemical and biological processes.
The main processes include the displacement of soil Hg(0) by water, the "wick effect" to increase the
photo-reducible Hg(II) pool in surface soils, and the increased soil moisture to promote the biotic and
abiotic reduction of Hg(II). Another well-documented process is that the atmospheric wet deposition of
Hg will increase the Hg pool in surface soils and the newly deposited Hg is very active to reduce to
Hg(0) (Hintelmann et al., 2002), although our study did not focus on this issue. During June 2014
campaign, no precipitation occurred in the first two days (6–7 June 2014), but the rest days were
rainy/snowy days (Fig. 2). We tried to use the daily Hg(0) flux of the two sunny days as the benchmark
to compare with those of rainy/snowy days to investigate the effect of precipitation on the Hg(0) flux
over the timescale of one day. Figure 6 showed that the daily Hg(0) flux for rainy/snowy days were
higher (ranging from 16% to 154%) than the mean of the two sunny days. The result indicates that the
precipitation increased soil Hg(0) emission on the timescale of one day, although the low solar radiation
and temperature on rainy/snowy days would potentially decrease soil Hg(0) emission, as mentioned
above.
**3.3.3 Effect of solar radiation and soil temperature on air–surface Hg(0) flux**
Almost all laboratory experiments and field measurements, including this study, show that the high
solar radiation and elevated soil temperature synergistically facilitate the soil Hg(0) emission (Edwards
and Howard, 2013; Park et al., 2014). The following hypotheses have been proposed to explain the role
of solar radiation and temperature in promoting soil Hg(0) emission, including (1) solar radiation
promotes the photo-reduction of Hg(II) in surface soils to form Hg(0) in short time scale; (2) solar
radiation and high soil temperature reduce the apparent activation energy of Hg(0) desorption and
increase Hg(0) emission from surface soils; and (3) the high soil temperature favors the Hg(0)
production in soil column by biotic and abiotic processes (Carpi and Lindberg, 1998; Gustin et al.,

2002).





379  Many studies used the Arrhenius equation (Eq. 2) to quantitatively investigate the relationship

380 between soil temperature and Hg(0) flux.

381   $$F = Ae^{-Ea/RT} \text{ or } \ln(F) = \ln(A) - \frac{Ea}{RT} \quad \text{(Eq. 2)}$$

382 where $F$ is the Hg(0) flux (ng m$^{-2}$ h$^{-1}$), $R$ is the gas constant, $T$ is the soil temperature (K), $A$ is the pre-

383 exponential factor and $Ea$ is the apparent activation energy. A plot of $\ln(F)$ versus $1/T$ obtains a straight

384 line with intercept equal to the log of the $A$, and the slop equal to $-Ea/R$. Theoretically, the concept of

385 the apparent $Ea$ refers to the thermally controlled reaction. Therefore, Hg(0) flux induced by light and

386 precipitation should be excluded from the correlation analysis. However, in many previous studies,

387 especially for the field measurements, the bulk Hg(0) flux in the light was generally used to explore the

388 contribution of solar radiation or temperature to Hg(0) flux and did not isolate the respective effect of

389 the two factors (Fu et al., 2008a). This will systematically overestimate or underestimate the

390 contribution of solar radiation and temperature on soil Hg(0) emission depending on the source or sink

391 of soil for Hg(0) in the dark. Only in some controlled experiments, the separated data was used to

392 explore the respective role of radiation and temperature in soil Hg(0) emission (e.g., Kocman and

393 Horvat, 2010).

394  In this study, for respectively determining the contribution of solar radiation and temperature on

395 the Hg(0) flux, besides Hg(0) flux was measured in the natural light, Hg(0) flux in the dark was also

396 measured simultaneously with a foil-covered chamber. The temperature-corrected Hg(0) flux (i.e., bulk

397 Hg(0) flux in the light – Hg(0) flux in the dark) in daytime (PAR > 0) was considered to be the

398 contribution of the solar radiation. As mentioned above, the effect of precipitation should be excluded

399 from the data set, therefore we only collected Hg(0) flux data on days without precipitation during

400 December 2014 and May–June 2015 campaign.

401  Figure 7 displays the temporal variation of bulk Hg(0) flux in the light, Hg(0) flux in the dark, net

402 Hg(0) flux in the light (i.e., bulk Hg(0) flux in the light – Hg(0) flux in the dark) and the environmental

403 variables. Obviously, changes in solar radiation had a greater influence on soil Hg(0) flux than did

404 changes in soil temperature. The data showed that the soil served as a Hg(0) sink during all study days





in December 2014 campaign in the dark with high deposition flux in low soil temperature and low
deposition flux in high soil temperature. During study days of May–June 2015 campaign, the soils
served as a very low Hg(0) source in the midday with relatively high soil temperature. This finding is
consistent with many studies in background soils (Ericksen et al., 2006; Gustin et al., 2006; Fu et al.,
2008a; Edwards and Howard, 2013). It indicates that the soil temperature plays an important role in
Hg(0) dynamics at the air–soil interface, i.e., low soil temperature favors to absorb Hg(0) or reduce
Hg(0) emission.
After the temperature corrected, except for the midday of study days during May–June campaign,
the net Hg(0) flux in the light was higher than the bulk Hg(0) flux. The positive linear correlation was
found between cumulative PAR and cumulative Hg(0) flux in the daytime, although cumulative PAR
only explained ~28% of variation in cumulative Hg(0) flux in the daytime (Fig. S2 in Supplement).
We used the Hg(0) emission data set in the dark to calculate the $Ea$ using the Arrhenius equation.
Since the soils in the dark was the sink of atmospheric Hg(0) in most of the study period, only limited
data set (n=9) can be used (Fig. S3 in Supplement). The $Ea/R$ for Hg(0) emission from our remote soils
with extremely low Hg concentrations (~12 ug kg$^{-1}$) was 30.40. Table S1 in Supplement lists the $Ea/R$
for different soils with large variation of soil Hg concentrations, including this study, and shows that the
$Ea/R$ of Hg(0) emission from soils with low Hg concentrations was higher than those of soils with high
Hg concentrations and significantly lower than that of theoretical value (7.31) of elemental Hg. It
indicates that surface soils with high Hg concentrations has great potential to emit Hg(0). This trend is
consistent with the laboratory study of Bahlmann et al. (2006), although the availability of Hg in soils
also significantly regulate Hg(0) emission (Bahlmann et al., 2006; Kocman and Horvat, 2010).
The QTP is characterized by high solar radiation with intense UV radiation. We further performed
the controlled experiment to quantify the role of different wavebands of solar radiation (UVB, UVA and
visible light) in Hg(0) flux. Figure 8 shows that UV radiation was the dominant waveband of solar
radiation for Hg(0) emission in the daytime, contributing >80% of Hg(0) emission in the light, and the
contribution of UV-B radiation accounted for >50% in all study days. This finding is consistent with
previous laboratory studies (Moore and Carpi, 2005; Bahlmann et al., 2006; Xin et al., 2007).



## 4 Conclusions and implication

In this study, we measured the Hg(0) flux between the air and surface permafrost soil in the QTP. We also performed the controlled field experiments to explore the effect of precipitation and different wavebands of solar radiation on the Hg(0) exchange air–soil interface. The result showed that the environmental conditions, including solar radiation, soil temperature and precipitation, greatly influenced the Hg(0) exchange between air and surface.

This study and other field measurements and laboratory experiments have clarified that the fate and transport of soil Hg is very sensitive to the environmental variables (Krabbenhoft and Sunderland, 2013). Therefore, our results have several important implications to the Hg biogeochemical cycle in the soils of QTP under the rapid climate warming and environmental change. Firstly, the increased surface temperature in the QTP will potentially promote the remobilization of soil Hg. Field measurements and modeling study have revealed that the surface temperature in the QTP is increasing, and the warming trend exceeds those for the Northern Hemisphere and the same latitudinal zone (Kang et al., 2010). Secondly, the increased UV radiation in the QTP may improve Hg(0) emission from surface soils. UV radiation reaching the surface of the QTP is estimated to increase because of the decrease of stratospheric $O_3$ (Zhou et al., 2013). Our result and many above-mentioned studies show that UV radiation plays the primary role in promoting the surface Hg(0) emission in the daytime. Thirdly, the temp-spatial pattern of precipitation in the QTP is also altering (Kang et al., 2010), which potentially alters the flux and temp-spatial pattern of air–soil Hg(0) exchange in this region because of the importance of precipitation on the Hg(0) exchange. However, this study was just the beginning to explore the effect of climate change on the terrestrial Hg cycle in the QTP. The large uncertainties highlight that more researches are needed in the future.

## Acknowledgements

The study was financially supported by the National Key Basic Research Program of China (No. 2013CB430002), National Natural Science Foundation of China (Nos. 41573117, 41371461, 41203068), and Young Scientists Fund of Research Center for Eco-Environmental Sciences, Chinese Academy of Sciences (No. RCEES-QN-20130048F). We thank the staff of the Beiluhe Permafrost Engineering and





Environmental Research Station affiliated to the Cold and Arid Regions Environmental and
Engineering Research Institute, Chinese Academy of Sciences (CAREER–CAS) for their assistance.

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





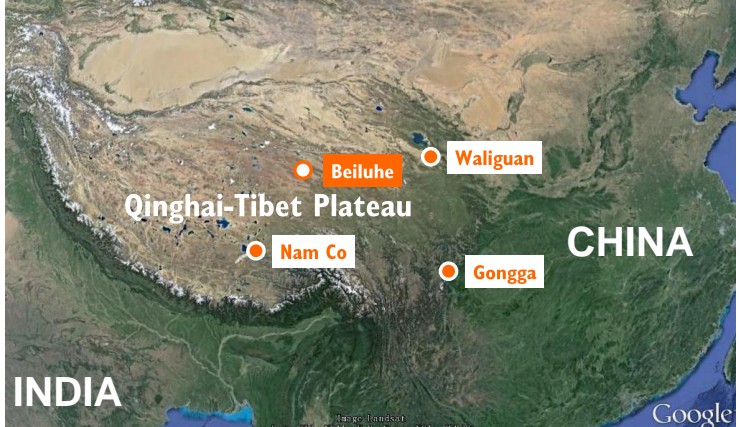



Figure 1. Locations of the Beiluhe station (4760 m a.s.l., this study), Mt. Waliguan (3816 m a.s.l., Fu et al., 2012), Mt.
Gongga (1640 m a.s.l., Fu et al., 2008b) and Nam Co (4730 m a.s.l., Yin et al., 2015) where atmospheric Hg were
determined.







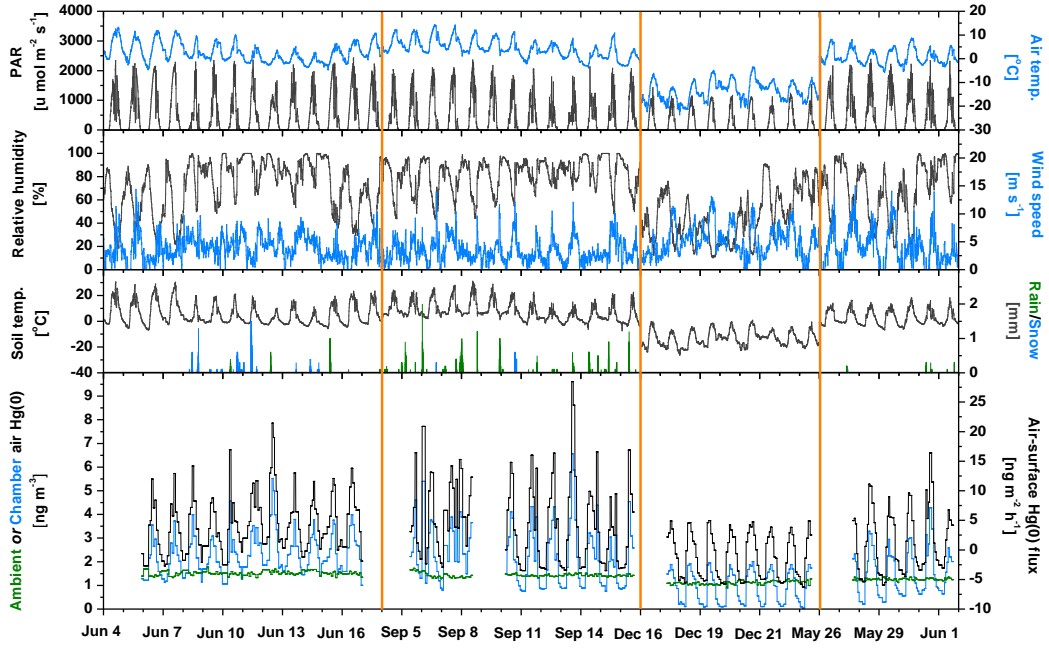



Figure 2. Temporal variation of environmental variables, air Hg(0) concentrations inside and outside of chamber, and air–surface Hg(0) flux at the Beiluhe station in the central QTP during four campaigns in 2014–2015.










Figure 3. Seasonal variation of Hg(0) concentration in ambient air and air–surface Hg(0) flux during four campaigns at the
Beiluhe station in the central QTP in 2014–2015.





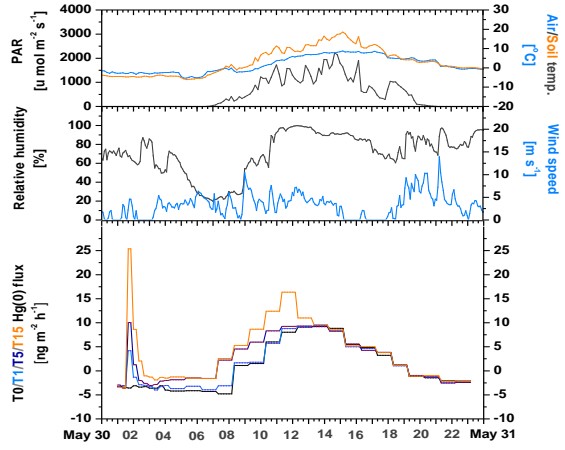



Figure 4. Temporal variation of Hg(0) flux over four soil plots with different treatment of water addition (T0: 0 mm
treatment, T1: 1 mm treatment, T5: 5 mm treatment and T15: 15 mm treatment) and the environmental variables.












Figure 5. Increased Hg(0) emission for three different treatments (1 mm, 5 mm and 15 mm addition of water) compared with
the 0 mm treatment during the controlled experiment on 30 May 2015.





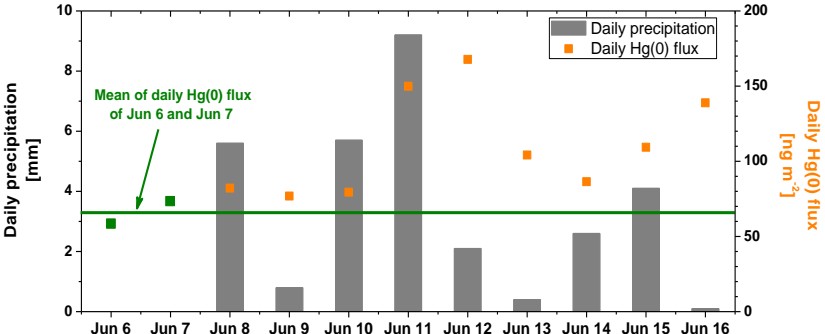



Figure 6. Daily Hg(0) flux and daily precipitation in June 2014 campaign.


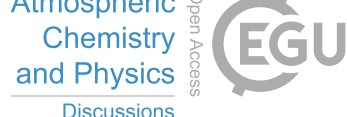

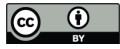



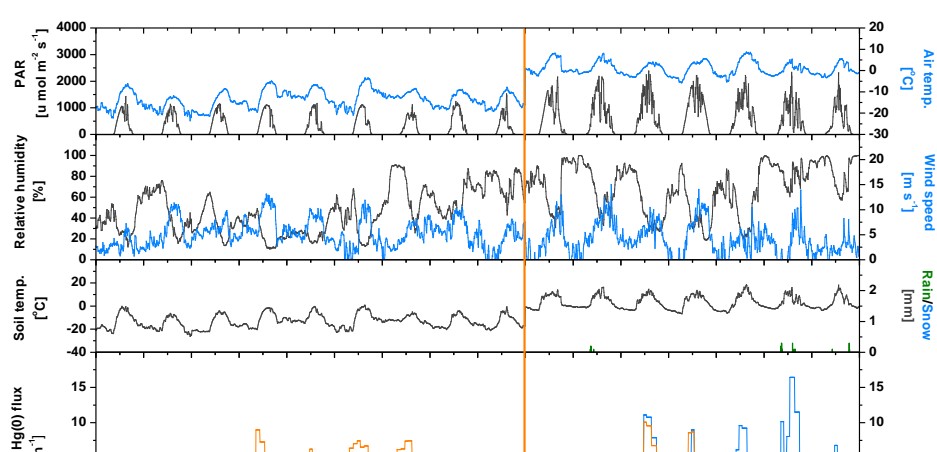



Figure 7. Temporal variation of bulk Hg(0) flux in the light, Hg(0) flux in the dark, and net Hg(0) flux in the light (bulk Hg(0) flux in the light–Hg(0) flux in the dark) in six study days without precipitation during December 2014 campaign and May–June 2015 campaign.








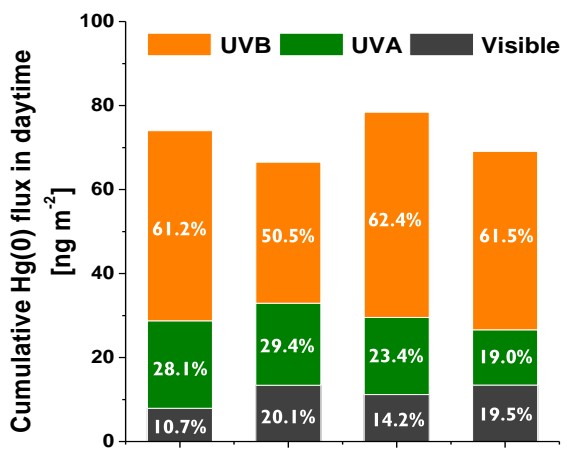



Figure 8. Cumulative Hg(0) emission flux in daytime triggered by UVB, UVA and visible light in four study days during

December 2014 campaign and May–June 2015 campaign. Four chambers with different exposure treatments were used to

measure Hg(0) flux simultaneously in the daytime. Chamber-A was used to measure the Hg(0) flux in the natural light.

Chamber-B and Chamber-C were covered with UVB filter and UV filters to remove the corresponding wavebands from the

natural light, respectively. Chamber-D covered with foil was used to measure Hg(0) flux in the dark. The experiments were

performed in four days without precipitation (21–22 December 2014 and 29–30 May 2015) to exclude the effect of

precipitation. Hg(0) flux triggered by UVB, UVA and visible light was equal to difference of flux between Chamber-A and

Chamber-B, between Chamber-B and Chamber-C, and between Chamber-C and Chamber-D, respectively. The transmittance

of UVB filter and UV filter was shown in Fig. S1 in Supplement.