# Peer review of "Air-surface exchange of gaseous mercury over permafrost soil: an"

_Atmospheric Chemistry and Physics, 2016_

## Referee Comment (RC1) · Anonymous Referee #2 · 12 Oct 2016

The manuscript Âż Air–surface exchange of gaseous mercury over permafrost soil: an investigation at a high-altitude (4700 m a.s.l.) and remote site in the central Qinghai-Tibet PlateauÂń by Ci et al. brings important new information about air-surface exchange patterns and mechanisms in a very specific environment for which such information is missing in scientific literature. In the light of changing environment and future global Hg cycling, this information is of paramount importance.

General comment: In general, the manuscript is well structured, information properly presented and appropriate conclusions drawn. As such, I believe it merits the criteria

to be published in ACP.

Here are some specific suggestions that might help to improve and strengthen the clarity of this paper: - Abstract: some numbers should be included in the abstract, e.g. about the magnitude of fluxes etc. - Line 23: What is relatively long timescale? Try to be more specific. - Line 31: What are favorable conditions? Perhaps first part of the sentence should be removed, as these conditions are discussed in detail later on. - Lines 93-108: This part is too general and should be significantly shortened or completely removed. - Line 144: Be more specific about soil plot/lithologic unit studied. - Lines 235-241: This part is too general and should be shortened or moved to the Introduction. - Lines 241-248: This part belongs to section 2.1 - Lines 270-283: This part belongs to the Method section. - Lines 452-453: What exactly you mean with "large uncertainties". Perhaps you should elaborate a bit more on this. Technical/linguistic comments: - Line 227: Remove "in" before "below". - Line 254: I suggest replacing "Investigators supposed..." with "Previous studies..." or similar. - Lines 249-253: I suggest rephrasing and combining this information in one sentence. - Line 445: Replace "improve" with "increase" or "enhance". - Figure 1: scale should be included.

---

## Referee Comment (RC2) · Anonymous Referee #3 · 9 Nov 2016

This manuscript investigated the mercury emission from permafrost soil in QTP and studied its controlling factors including the rainfall, snowfall, soil temperature and solar radiation. This work is very significant for this region with unique climate condition. I recommend this paper to be accepted. In addition, more studies were needed to explore its mechanism 1. Line 99, please give the specific flushing flow rate. 2. Line 410, low soil temperature is unfavorable for Hg(0) emission, however, how could understand your explanation of low soil temperature favors to absorb Hg(0).

---

## Author Response (AR1)

**We thank the reviewer for the insightful comments and valuable suggestions. We have incorporated the reviewer's suggestions into the revised manuscript to improve the quality of our paper. Please find our point-by-point responses to the comments below in bold.**

The manuscript "Air-surface exchange of gaseous mercury over permafrost soil: an investigation at a high-altitude (4700 m a.s.l.) and remote site in the central Qinghai-

[Figure]

Tibet Plateau" by Ci et al. brings important new information about air-surface exchange patterns and mechanisms in a very specific environment for which such information is missing in scientific literature. In the light of changing environment and future global Hg cycling, this information is of paramount importance.

General comments: In general, the manuscript is well structured, information properly presented and appropriate conclusions drawn. As such, I believe it merits the criteria to be published in ACP.
**Response: We appreciate the reviewer's recognition of the merits of this work.**

Here are some specific suggestions that might help to improve and strengthen the clarity of this paper:

- Abstract: some numbers should be included in the abstract, e.g. about the magnitude of fluxes etc.
**Response: We have followed the reviewer's suggestion to add the related data and numbers in the Abstract section.**

- Line 23: What is relatively long timescale? Try to be more specific.
**Response: The timescales have been added in the revised manuscript.**

- Line 31: What are favorable conditions? Perhaps first part of the sentence should be removed, as these conditions are discussed in detail later on.
**Response: We have followed the reviewer's suggestion to remove "Under favorable conditions".**

- Lines 93-108: This part is too general and should be significantly shortened or completely removed.
**Response: We have followed the reviewer's suggestion to greatly shorten this part.**

- Line 144: Be more specific about soil plot/lithologic unit studied.
**Response: The details of soil property and related reference have been added in**

[Figure]

**the revised manuscript.**

- Lines 235-241: This part is too general and should be shortened or moved to the Introduction.
**Response: We have followed the reviewer's suggestion to move this part to the Introduction section.**

- Lines 241-248: This part belongs to section 2.1
**Response: We have followed the reviewer's suggestion to move this part to Section 2.1.**

- Lines 270-283: This part belongs to the Method section.
**Response: Thank you for the constructive suggestion. A new sub-section (Section 2.3: Controlled field experiments) has been added into the Method section in the revised manuscript. We have moved the description of controlled field experiments about water addition and different waveband of solar radiation to Section 2.3.1 and 2.3.2, respectively.**

- Lines 452-453: What exactly you mean with "large uncertainties". Perhaps you should elaborate a bit more on this.
**Response: The sentence has been rewritten in the revised manuscript.**

Technical/linguistic comments:

- Line 227: Remove "in" before "below".
**Response: Done.**

- Line 254: I suggest replacing "Investigators supposed: : :" with "Previous studies: : :" or similar.
**Response: Done.**

-Lines 249-253: I suggest rephrasing and combining this information in one sentence.
**Response: Done.**

[Figure]

-Line 445: Replace "improve" with "increase" or "enhance".
**Response: Done.**

Figure 1: scale should be included.
**Response: Done.**

[Figure]

Atmos. Chem. Phys. Discuss.,
doi:10.5194/acp-2016-515-AC2, 2016

[Figure]
**We thank the reviewer for the insightful comments and valuable suggestions. We have incorporated the reviewer's suggestions into the revised manuscript to improve the quality of our paper. Please find our point-by-point responses to the comments below in bold.**

This manuscript investigated the mercury emission from permafrost soil in QTP and studied its controlling factors including the rainfall, snowfall, soil temperature and solar

[Figure]

radiation. This work is very significant for this region with unique climate condition. I recommend this paper to be accepted. In addition, more studies were needed to explore its mechanism.
**Response: We appreciate the reviewer's recognition of the merits of this work.**

Line 99, please give the specific flushing flow rate.
**Response: The specific flushing flow rate was given in the third paragraph of Section 2.2.**

2. Line 410, low soil temperature is unfavorable for Hg(0) emission, however, how could understand your explanation of low soil temperature favors to absorb Hg(0).
**Response: This issue has been addressed by the previous studies (e.g., Park et al., 2014 and references therein). The related reference has been added in the revised manuscript.**
* * *
[Figure]

**Air–surface exchange of gaseous mercury over permafrost soil: an investigation at a high-altitude (4700 m a.s.l.) and remote site in the central Qinghai-Tibet Plateau**

Zhijia Ci[1], Fei Peng[2], Xian Xue[2], and Xiaoshan Zhang[1]

[1]Research Center for Eco-Environmental Sciences, Chinese Academy of Sciences, Beijing, 100085, China

[2]Cold and Arid Regions Environmental and Engineering Research Institute, Chinese Academy of Sciences, Lanzhou, 730000, China

*Correspondence to*: Z. J. Ci (zjci@rcees.ac.cn)

**Abstract.** The pattern of air–surface gaseous mercury (mainly Hg(0)) exchange in the Qinghai-Tibet Plateau (QTP) may be unique because this region is characterized by low temperature, great temperature variation, intensive solar radiation, and pronounced freeze-thaw process of permafrost soils. However, air–surface Hg(0) flux in the QTP is poorly investigated. In this study, we performed field measurements and controlled field experiments with dynamic flux chambers technique to examine the flux, temporal variation and influencing factors of air–surface Hg(0) exchange at a high-altitude (4700 m a.s.l.) and remote site in the central QTP. The results of field measurements showed that surface soils were net emission source of Hg(0) in the entire study (2.86 ng m$^{-2}$ h$^{-1}$ or 25.05 μg m$^{-2}$ y$^{-1}$). Hg(0) flux showed remarkable seasonality with net high emission in the warm campaigns (June 2014: 4.95 ng m$^{-2}$ h$^{-1}$; September 2014: 5.16 ng m$^{-2}$ h$^{-1}$ and May–June 2015: 1.95 ng m$^{-2}$ h$^{-1}$) and net low deposition in winter campaign (December 2014: –0.62 ng m$^{-2}$ h$^{-1}$), 
[revised manuscript text omitted]